# Does Civic Engagement Support Relational and Mental Health of Urban Population?

Michal Hrivnák [1,*], Peter Moritz [1], Katarína Melichová [1] and Soňa Bellérová [2]

1   Institute of Regional and Rural Development, Faculty of European Studies and Regional Development, Slovak University of Agriculture in Nitra, Tr. A. Hlinku 2, 949 76 Nitra, Slovakia

2   Institute of Landscape Architecture, Faculty of Horticulture and Landscape Engineering, Slovak University of Agriculture in Nitra, Tr. A. Hlinku 2, 949 76 Nitra, Slovakia

*   Correspondence: xhrivnak@uniag.sk

**Abstract:** There is a general assumption that there is a relationship between civic engagement and mental health, but it has still received limited attention in empirical studies. This study provides estimates of the impact of civic engagement (measured in terms of political and community engagement) on the health of individuals in the case of a medium-sized urban settlement within the context of a post-socialist country. The impacts of civic engagement on mental and relational health are distinguished, which have received little attention in studies on the topic. Using primary data and utilising the tools of econometrics, we found positive effects of the population's community engagement, including positive effects of volunteering, on relational health. Political participation of the population contributed to the reduction of depressive symptoms, but the relationship between community engagement and mental health was not found. A relatively high propensity towards participation in health and well-being projects, leading to improvements in the collective approach to public health and addressing unhealthy conditions in communities, was identified in the sample.

**Keywords:** community engagement; political engagement; volunteerism; well-being; mental health; relational health

## 1. Introduction

Civic engagement, including political engagement and community engagement, is expected to have a mostly positive impact on health [1–3]. Activities of communities and social movements can have direct effects on public health via successful implementation of projects initiated by active communities (e.g., health and well-being projects, solutions of environmentalists) and social movements addressing health issues, e.g., HIV/AIDS, or immunisation campaigns [4]. However, there are many more arguments for investigating the relationship between civic engagement and health. Civic engagement is potentially an important intermediary for raising the collective responsibility for health equity [5] and addressing unhealthy conditions in communities [6]. Via engagement in policymaking, individuals can affect the emergence of policies aimed to deliver solutions improving public health [7]. However, as suggested by Chau [8], it is difficult to measure and compare these impacts as the level of civic engagement in policy making varies from informing the public to full community engagement.

Our intention in this empirical study is to estimate the impact of civic engagement, both in terms of political engagement and community engagement, on the health of individuals in the case of smaller, medium-sized urban settlements. By setting up this main goal, we would like to address several gaps in the knowledge. First of all, we still do not have sufficient empirical evidence on the effects of community engagement on relational and mental health [6]. The number of studies focusing on the impact of certain types of civic engagement on health (best represented by volunteering) is growing (see, e.g., [9]); however, far fewer studies try to find a framework for the complex measurement of civic, social

or political engagement impacts. Most of the available empirical evidence evaluates the relationship between civic engagement and health on samples of respondents within a wider space or metropolitan areas [10]; however, several authors point to the fact that the level of civic engagement is rather higher in smaller and medium-sized cities than in large and metropolitan cities [11,12]. Moreover, the majority of studies focus rather on estimating the relationship between civic engagement and mental health [3,9,13–15]. Taking into account the fact that the ability to generate healthy interpersonal bonds is one of the main prerequisites for the reduction of depressive symptoms, relational health can at most be considered a subset of mental health [16], while we assume that different forms of civic engagement can have different effects on mental and especially relational health. At the same time, based on our best knowledge, we do not know of any publication on the topic in the context of the Slovak Republic.

The city of Nitra meets our requirements for research on the effects of civic engagement in a small, medium-sized city (with a population size of 76,932). At the same time, despite the fact that it is a regional center of the NUTS III region, Nitra is classified as a rural city in view of the rurality criteria based on population density or commuting distances to work [17], which suits us in view of the assumption of multiple links bridging social capital and civic engagement in smaller cities [12]. At the time of the survey, the local government was composed of activists for 4 years, which, according to some authors (e.g., [18]), should contribute to the creation of a suitable foundation for the development of civic engagement activities. Since part of the studies point to a potential relationship between public health and work for the community (e.g., [19,20]), we also want to compare the effects of aggregated indicators of "community" and "political" engagement on volunteering measured directly by average volume of hours of volunteer work against the relational and mental health of individuals. Our measurements will be focused on a small, medium-sized city in the conditions of a post-socialist country. The empirical study can be understood as a case study of the city of Nitra, participating in the Horizon 2020 "IN-HABIT" project in which the impacts of co-design and co-deployment of experimental solutions in public spaces on the health and well-being of the population are measured and evaluated. Thus, the study represents a baseline analysis.

## 2. Theory Background

The focus of our study is on three concepts, the interrelationship of which is the subject of our investigation. These three concepts are civic engagement, mental health and the relational health.

Civic engagement is a broad term usually referring to collective action carried out with a socially-oriented interest, integrating, e.g., volunteering, community engagement, social and political participation [21]. Civic engagement can be understood as set of individual efforts aimed to make a difference in society and communities while developing the knowledge, skills, values and motivation necessary to make that difference [22]. As this means that civic engagement promotes the quality of life in a community both through political and social processes, we will further consider social (or more specifically, community) and political engagement as subsets of civic engagement.

Mental health is perceived and measured differently in the literature and lacks a firm conceptual definition [23]. One can observe a very wide use of this term as a euphemism for mental disorders [24]. In line with Bhugra and Sartorius [25], we define mental health in a broader sense as a state of individuals, including biological, psychological or social factors, that contribute to a person's mental state and ability to function within the environment. Relational health can be understood trough interpersonal interactions that are growth-fostering or mutually empathic and empowering. Poor relational health increases an individual's risk of developing psychological distress [26]. This term was first used by Liang et al. [27] with intention to develop an instrument designed specifically to assess relational health. Relational health is a developed state of health and connectedness in which a person has the opportunity and capacity to:

- Develop, maintain, receive and perceive safe, stable and nurturing relationships with other individuals;
- Participate in a broad range of social relationships including active engagement in a variety of social activities and social support.

Empirical studies paid more attention to the relationship between civic engagement and well-being than mental or relational health [21,28–30]. Well-being is a complex multidisciplinary concept that has its physical, psychological and socioeconomic contexts, which, however, means that as a construct it has the potential to unify the needs of several sectors and scientific disciplines and has certain overlaps with relational and mental health [31]. Marciniak et al. [30] investigated the relationship between psychological well-being and civic engagement during COVID-19 on a sample of 1362 academic students from Poland, Lithuania and Croatia. They found that the students with higher levels of psychological well-being reach higher levels of civic engagement. Wray-Lake et al. [21] found a positive relationship between volunteering, pro-environmental behaviour and feelings of happiness and life satisfaction. Jenkinson et al. [9] proved that an individualistic and consumption-oriented lifestyle had negative impact on the well-being of individuals.

Civic engagement, and especially community engagement, can affect not only life-satisfaction but also the health of individuals [6]. To better understand these potential links, it is proper to state that almost 50% of health outcomes are to some extent predetermined by socioeconomic determinants. In the context of the US, Artiga and Hinton [3] found that health outcomes are driven by an array of factors, including underlying genetics, health behaviours, social and environmental factors and health care. Health care directly affects, on average, less than 20% of health outcomes. Public health is also affected by factors of economic stability, factors pertaining to the physical space of residences and neighbourhoods, and those pertaining to education or institutions within the communities in which the individual participates [32].

Several studies investigated the relationship between civic engagement and mental health. This relationship is often expected to be bi-directional, while more studies found the impact of higher levels of health and well-being on future social or community engagement. The results of Fang et al. [15] indicated consistent cross-lagged associations from higher well-being to higher future civic engagement. However, they found no reverse relationship. On the other hand, Wray-Lake et al. [21] also found that adolescents and early young adults' community engagement, measured both in terms of social and political engagement, contributed to decreases in later depressive symptoms. Landstedt et al. [13], in a longitudinal study of 1001 individuals, found that in the case of men, involvement in social networks promotes health, most likely through the provision of social and psychological support, perceived influence and sense of belonging. However, they found no relationship in the case of women. The role of community engagement in the improvement of public health is also stressed in the results of Guo et al. [14], as they proved that in the case of China, association between civic engagement and happiness is moderated by income. Civic engagement is more beneficial for low- and high-income people than for the middle-class population. Concerning impacts of volunteering on mental health, Jenkinson et al. [9] found that volunteering had a strong positive impact on both reducing depression and supporting well-being.

Few studies have yet focused on evaluating the relationship between civic engagement and relational health, or the quality and intensity of relationships that an individual generates in different environments. Dang et al. [33] found that individuals with a stronger connection to a neighbourhood demonstrated a greater willingness to undertake voluntary actions and engage in community projects. Rubin et al. [34] found a strong role of relationship quality with parents, peers and friends, as well as of socially competent behaviours, in the development of civic competences. Due to low stock of literature on the topic of social and political (civic) engagement effects on self-perceived relational and mental health, we put forward following research hypotheses:

**H1.** *Growth in the level of community engagement has a positive effect on the relational health of individuals.*

**H2.** *Growth in the level of community engagement has a positive effect on the metal health of individuals.*

**H3.** *Growth in the level of political engagement has a positive effect on the relational health of individuals.*

**H4.** *Growth in the level of political engagement has a positive effect on the mental health of individuals.*

## 3. Materials and Methods

The relationship between civic engagement and both relational and mental health will be investigated using primary data obtained through an electronic survey conducted in the context of the selected small, medium-sized city of Nitra in the context of the post-socialist economy of Slovakia. Within the chapter, we will first characterise the primary data used and the design of the unique survey, and then we will present the research framework for examining the outlined relationships.

### 3.1. The Survey

Our investigation is based on the utilisation of the dataset which results from the survey "Well-being of the population, health and active citizenship in the city of Nitra", which was conducted in 2022 due to the needs of the Horizon 2020 project "IN-HABIT". The population is represented by 76,932 citizens of the city of Nitra (01.01.2022, according to the Statistical Office of the Slovak Republic) and the obtained sample of 318 respondents. We consider the obtained sample to be representative, despite the fact that at a confidence level of 95%, the margin of error in the case of a sample of the given size slightly exceeds the limit of 5 within the rule of thumb, reaching the level of 5.28, which is still satisfactory. As many as 7 questionnaires were discarded due to the incorrectness of part of the answers and highly influential, unrealistic values. The survey was carried out in electronic form. The population of the city was reached by sharing the questionnaire in groups on social networks. Data collection was carried out in two stages. In the first stage, the questionnaire was shared in 29 public groups on the Facebook social network. These 29 groups represent all groups on the social network that were created for the purpose of discussing local challenges in the conditions of individual residential zones in the city. At the same time, it is necessary to state that this social network has a highly dominant position in the conditions of Slovakia [35], even in the case of older age categories of the population. These 29 groups reached a total of 52,432 followers. Through sharing, survey respondents invited other residents in the city to participate in the survey. Thus, the so-called snowball-sampling method was employed. This technique of data collection is used when it is difficult to access subjects with the target characteristics (in our case—residents of Nitra). In such a case, the participants of the survey recruit other subjects within their social ties until the saturation of the survey is reached [36]. Only residents of Nitra participated in the survey. The questionnaire consisted of 32 questions of which 20 were closed or semi-closed questions, 8 were open questions and 4 were matrices (checkbox grids). The construction of the key 4 aggregated variables, which were collected in the survey through the aforementioned 4 matrices, necessarily requires justification. These variables are: (1) relational health, (2) mental health, (3) political engagement and (4) community engagement.

Our aggregated variable for measuring relational health represents a simplified and adapted version of two of the three components of the RHI measure (relational health indices) developed by Liang et al. [27]. Specifically, these are the relationship to mentors and community components within RHI, which were transformed into a simple 10-question framework. The list of 10 questions to measure the relational health is enclosed as Appendix A. The variable "mental health" can be also understood as a measurement of depressive symptoms and is based on the CESD-10 framework adapted by Kashcheeva [37].

This framework is enclosed as Appendix B. Concerning the indicators "political engagement", understood in terms of local political engagement and "community engagement", there are uniquely constructed based on literature [6,38–40], while the questions for aggregated indicators have been tested using the expert review method [41]. The initial framework of questions was tested by 25 experts with expertise in the fields of civic engagement and civic participation, while 10 experts were university scientists, 5 were managers of participatory processes within public administration bodies and 7 were managers of NGOs focusing on the development of public planning and civic engagement. The resulting frameworks represent a modification after the incorporation of the majority of suggestions and comments. The testing experts were invited to score the importance of the given factors on a Likert scale, to suggest the exclusion of a factor, to modify the formulation of the question to a particular factor or to suggest an alternative factor. Frameworks of factors for these aggregated indicators are enclosed as Appendix A.

*3.2. Sample Obtained*

The survey was suspended at the time when it was no longer possible to obtain additional responses by repeated sharing, and when the obtained sample could be considered relatively balanced. First of all, it was necessary to obtain a sample that replicates the distribution of the population according to individual city districts, which was achieved. As of 01.01.2022, almost 60% of the city's residents lived in the three largest districts, while these districts (Old Town, Klokočina neighbourhood and Chrenová neighbourhood) represented 54.7% of respondents in the sample.

The sample is relatively balanced in terms of the distribution of respondents by age as 30.8% of respondents were in the age category of 15–30 years and 63.4% of respondents were between 30–60 years of age. An imbalance was recorded in the case of the age category over 60 years where we managed to obtain only 5.8% of the total number of respondents, but this was probably influenced by the lower ICT skills of elderly people. Up to 48.2% of the sample were women and 51.8% were men, 47.9% were married and 52.1% were single. The proportion of persons with higher education in the sample is slightly increased in case of I, II and III degrees (up to 51.4% of respondents), which is probably due to the fact that the data was collected within a survey promoted by a university. We also recorded a relatively balanced sample in terms of income of the population, with adequate income variability around a reasonable wage (1298 euros), which in the survey was close to the average wage, according to the data of the Statistical Office of the Slovak Republic (1342 euros as of 01.01.2022).

*3.3. Research Design*

The dependent variables in our statistical model will be "relational health" and "mental health", which were constructed in the manner described above. The list of variables included in the model is summarized in Table 1.

We will test our hypotheses through a regression model using cross-sectional data for 311 respondents of this extensive survey. Considering the potential multicollinearity between the factors of community engagement, volunteerism and participation on local development projects, we will build 3 models examining the relationship between the aggregate variable "relational health" and predictors, and 3 models examining the relationship between the aggregate variable "mental health" and predictors. Two considered predictors were excluded from the composition of the model as a result of specification goodness-of-fit tests. The resulting model showed the problem of the presence of heteroscedasticity in the data, which was solved by robust estimates. The problem of multicollinearity was not noted as the VIF reached low values. Data were standardised using the z-scoring method. Our theoretical models are as follows:

1. **RELATIONAL HEALTH** = $\beta 1 \times AGE + \beta 2 \times GENDER + \beta 3 \times FAITH + \beta 4 \times FAMILY STATUS + \beta 5 \times CHILDREN + \beta 5 \times EDUCATION + \beta 7 \times INCOME + \beta 8 \times FREE TIME + \beta 9 \times POLITICAL\_ENG + \beta 10 \times SOCIAL\_ENG + \alpha i + \varepsilon it$;

2. **MENTAL HEALTH** = β1 × AGE + β2 × GENDER + β3 × FAITH + β4 × FAMILY STATUS + β5 × CHILDREN + β5 × EDUCATION + β7 × INCOME + β8 × FREE TIME+ β9 × POLITICAL_ENG + β10 × SOCIAL_ENG +αi + εit;

3. **RELATIONAL HEALTH** = β1 × *AGE* + β2 × *GENDER* + β3 × *FAITH* + β4 × *FAMILY STATUS* + β5 × *CHILDREN* + β5 × *EDUCATION* + β7 × *INCOME* + β8 × *FREE TIME*+ β9 × *VOLUNTEERSHIP* + αi + εit;

4. **MENTAL HEALTH** = β1 × AGE + β2 × GENDER + β3 × FAITH + β4 × FAMILY STATUS + β5 × CHILDREN + β5 × EDUCATION + β7 × INCOME + β8 × *FREE TIME*+ β9 × *VOLUNTEERSHIP* + αi + εit;

5. **RELATIONAL HEALTH** = β1 × *AGE* + β2 × *GENDER* + β3 × *FAITH* + β4 × *FAMILY STATUS* + β5 × *CHILDREN* + β5 × *EDUCATION* + β7 × *INCOME* + β8 × *FREE TIME*+ β9 × *PARTICIP_LDP* + αi + εit;

6. **MENTAL HEALTH** = β1 × AGE + β2 × GENDER + β3 × FAITH + β4 × FAMILY STATUS + β5 × CHILDREN + β5 × EDUCATION + β7 × INCOME + β8 × *FREE TIME*+ β9 × *PARTICIP_LDP* + αi + εit.

**Table 1.** Description of variables used to test outlined relationships.

| Name of Variable | Description of Variable | Variable Type |
|---|---|---|
| *AGE* | age of the respondent | nominal |
| *GENDER* | dummy expressing gender (1—man, 0—woman) | binary |
| *FAITH* | dummy expressing faith in god (1—man, 0—woman) | binary |
| *FAMILY STATUS* | dummy expressing marital status (1—married, 0—not) | binary |
| *CHILDREN* | absolute number of born children | nominal |
| *EDUCATION* | educational level expressed on Likert scale 0–5 | ordinal |
| *INCOME* | absolute volume of average nominal wage | nominal |
| *FREE TIME* | absolute number of average number of free time hours | nominal |
| *POLITICAL_ENG* * | aggregated indicator (10 q. framework) of pol. particip. level (0–5) | ordinal |
| *COMMUNITY_ENG* * | aggregated indicator (10 q. framework) of community eng. level (0–5) | ordinal |
| *VOLUNTEERSHIP* * | average monthly number of hours spent by volunteering | ordinal |
| *PARTICIP_LDP* * | freq. of participation on local dev. projects supp. health and well-being | ordinal |

* All questions related to these indicators were bound to activities carried out by individuals in recent 12 months.

## 4. Results

*4.1. Patterns of Relational and Mental Health within the Sample*

The relational and mental health of the population is influenced by a wide range of factors, among which socioeconomic determinants can also play a role. In this chapter, however, we first want to describe the basic patterns of relational and mental health in the sample and better understand the distribution of respondents in relational and mental health score intervals according to selected socioeconomic factors.

In Figure 1, we can see the shares of respondents in the intervals of the mental and relational health variables, which represent an aggregation based on the average of the 10-question frameworks. In the context of the observed small, medium-sized city, we identified a higher average level of relational health than mental health. This is also demonstrated by the fact that within the interval of the average score from 4.01 to 5.00 fall up to 34.08% of respondents in the case of the aggregated variable "relational health", and only 22.51% of respondents in the case of the "mental health" indicator. However, there does not seem to be a pattern of mutual dependence between relational and mental health in the sample, which justifies the need to evaluate them separately.

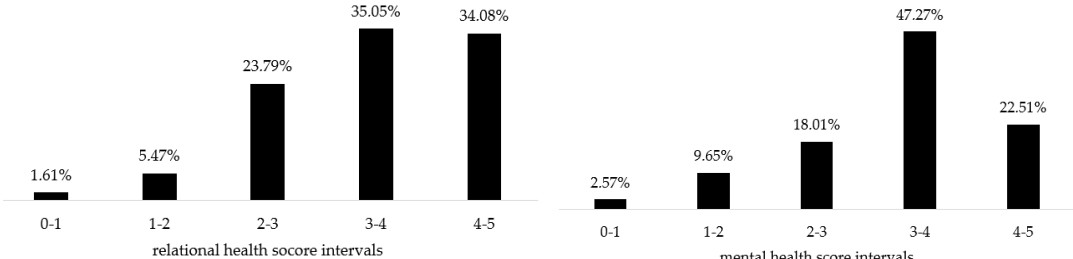

**Figure 1.** Comparison of respondents' distribution across intervals of relational and mental health.

The results for individual factors within the 10-question frameworks for our aggregated variables indicate which factors were key for higher or smaller identified levels of relational and mental health. The population showed the ability to generate healthy relational ties mainly through building ties with friends based on mutual trust (avg. score 4.04) and a high intensity of contact with their family (avg. score 3.71). On the contrary, the lowest average results in the evaluation of relational health were achieved in the case of investing a considerable amount of time in meeting new people (avg. score 2.45) and perceived belonging to different population groups and communities of interest (avg. score 2.45). The results of the survey showed that the average for the aggregate variable "mental health" was dragged down mainly by absenting feelings of optimism about the future (avg. score of 1.89) and by feelings of fear (avg. score 2.32). On the other hand, high-quality sleep (avg. score 3.67) and a low level of feelings of exertion during normal activities (avg. score 3.07) contributed to high mental health values.

In Tables 2 and 3, we compare the values of the total score of relational and mental health according to selected basic indicators—age, gender, level of education, family status and income level. The results indicate that the level of relational health decreases with increasing age. The highest levels of relational health were recorded in the 15–30 age group, and the lowest were recorded in the over 60 age group. At the same time, the descriptive results indicate that relational health is not differentiated depending on gender, income or family status. A certain growth of the relational health rate could be observed in the case of population groups with higher education.

**Table 2.** Share of population within intervals of relational health score by age, gender, education, marital status and level of income.

|  | 0–1 | 1–2 | 2–3 | 3–4 | 4–5 | Abs. Total |
|---|---|---|---|---|---|---|
| age 15–30 | 2.08% | 2.08% | 19.79% | 40.63% | 35.42% | 96 |
| age 31–45 | 2.33% | 4.65% | 20.16% | 37.98% | 34.88% | 129 |
| age 46–60 | 2.86% | 7.14% | 31.43% | 27.14% | 31.43% | 70 |
| age 60 and higher | 6.25% | 0.00% | 50.00% | 25.00% | 18.75% | 16 |
| women | 2.67% | 3.33% | 22.00% | 36.00% | 36.00% | 150 |
| men | 0.62% | 7.45% | 25.47% | 34.16% | 32.30% | 161 |
| elementary | 0.00% | 0.00% | 57.14% | 42.86% | 0.00% | 7 |
| secondary with certificate | 0.00% | 17.50% | 42.50% | 37.50% | 2.50% | 40 |
| secondary with graduation | 1.92% | 8.65% | 32.69% | 36.54% | 20.19% | 104 |
| tertiary of I. and II. grade | 2.34% | 0.78% | 13.28% | 33.59% | 50.00% | 128 |
| tertiary of III. grade | 0.00% | 0.00% | 9.38% | 31.25% | 59.38% | 32 |
| single | 0.67% | 7.38% | 22.82% | 36.91% | 32.21% | 149 |
| married | 2.47% | 3.70% | 24.69% | 33.33% | 35.80% | 162 |
| up to 500 eur | 3.33% | 3.33% | 33.33% | 40.00% | 20.00% | 30 |
| 500–750 eur | 4.26% | 8.51% | 40.43% | 23.40% | 23.40% | 47 |
| 750–1000 eur | 2.04% | 6.12% | 22.45% | 39.80% | 29.59% | 98 |
| 1000–1250 eur | 0.00% | 6.38% | 34.04% | 14.89% | 44.68% | 47 |
| 1250–1500 eur | 0.00% | 0.00% | 2.78% | 11.11% | 86.11% | 36 |
| 1500 and higher | 0.00% | 3.77% | 3.77% | 33.96% | 58.49% | 53 |

**Table 3.** Share of population within intervals of mental health score by age, gender, education, marital status and level of income.

|  | 0–1 | 1–2 | 2–3 | 3–4 | 4–5 | Abs. Total |
|---|---|---|---|---|---|---|
| 15–30 | 0.0% | 10.4% | 29.2% | 43.8% | 16.7% | 96 |
| 31–45 | 3.9% | 7.0% | 7.8% | 46.5% | 34.9% | 129 |
| 46–60 | 2.9% | 10.0% | 18.6% | 41.4% | 27.1% | 70 |
| 60 and higher | 6.3% | 25.0% | 31.3% | 25.0% | 12.5% | 16 |
| women | 4.7% | 11.3% | 22.7% | 46.7% | 14.7% | 150 |
| men | 0.6% | 8.1% | 13.7% | 47.8% | 29.8% | 161 |
| elementary | 0.0% | 0.0% | 71.4% | 28.6% | 0.0% | 7 |
| secondary with certificate | 7.5% | 12.5% | 32.5% | 32.5% | 15.0% | 40 |
| secondary with graduation | 1.0% | 10.6% | 54.8% | 16.3% | 17.3% | 104 |
| tertiary of I. and II. grade | 2.3% | 8.6% | 15.6% | 48.4% | 25.0% | 128 |
| tertiary of III. grade | 3.1% | 9.4% | 9.4% | 34.4% | 43.8% | 32 |
| single | 2.7% | 12.1% | 21.5% | 49.7% | 14.1% | 149 |
| married | 2.5% | 7.4% | 14.8% | 45.1% | 30.2% | 162 |
| do 500 eur | 10.0% | 10.0% | 23.3% | 40.0% | 16.7% | 30 |
| 500–750 eur | 4.3% | 17.0% | 29.8% | 31.9% | 17.0% | 47 |
| 750–1000 eur | 0.0% | 12.2% | 10.2% | 49.0% | 28.6% | 98 |
| 1000–1250 eur | 0.0% | 6.4% | 12.8% | 44.7% | 36.2% | 47 |
| 1250–1500 eur | 0.0% | 5.6% | 22.2% | 44.4% | 27.8% | 36 |
| 1500 and higher | 0.0% | 3.8% | 24.5% | 56.6% | 15.1% | 53 |

As for mental health, the age categories of 15–30 years and over 60 years appear to be more vulnerable as we recorded the highest average levels of mental health in the case of "middle-aged" age groups. Moreover, women appear to be slightly more vulnerable than men. A slight increase in average levels of mental health was also observed in the case of married people compared to single people, in the case of population groups with an income exceeding the average wage and also in the case of residents with a higher educational level (especially in the case of university graduates).

Thus, compared to relational health, mental health appears to be more selective depending on other sociodemographic or economic factors.

The data for the obtained sample also indicate a potential relationship between the population's political and social involvement and its relational and mental health. Certain relationships between selected factors of engagement in local development policy and health are demonstrated in Figure 2. Although we did not succeed in predicting a significant linear relationship between citizens´ passive participation in elections and relational or mental health, in the case of active participation (running for office), we already see a slightly positive influence for both relational and mental health. We consider the results for the impact of certain factors—namely, active participation in political discussions of online groups on social networks and participation in petitions, protests and boycotts—to be very interesting. Both of these factors showed a negative relationship with relational health and at the same time a positive relationship with mental health. We understand this result as an impulse towards in-depth research in behavioural sciences as it is probably influenced by the fact that citizens who are more active in the online space generate less voluminous and high-quality interpersonal ties, but it provides them with satisfaction and reduces feelings of depression. Similarly, an insurgent position toward local government decreases the likelihood of high levels of relational health. This result may be related to the growing degree of extremism within population in the context of Slovak cities and the fact that population groups with extreme attitudes form smaller and more excluded social groups. However, it is also true that a protest against local policy supports contributes to feelings of satisfaction and has a positive effect on mental health.

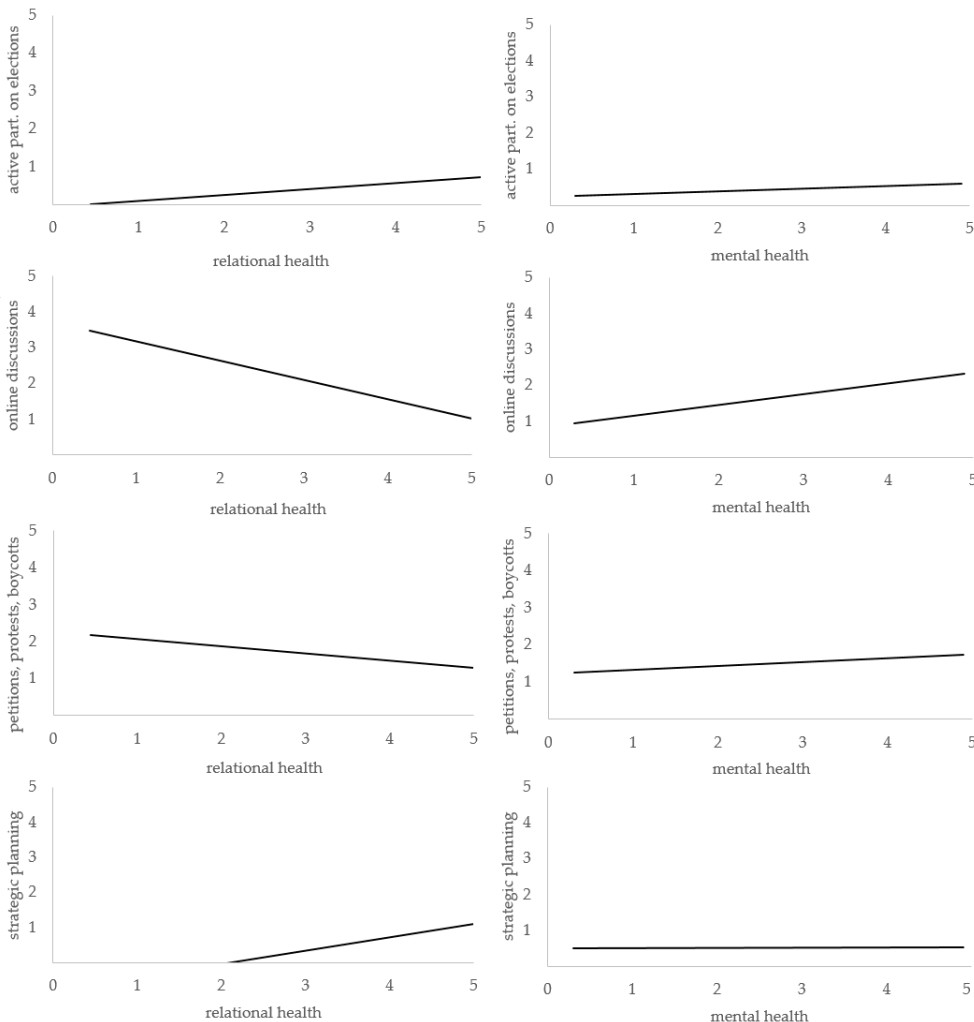

**Figure 2.** Prediction of linear relationships between several factors of political engagement, relational and mental health.

We also consider literature investigating the effects of participatory strategic planning on relational and mental health to be absent. Although the relationship between participation in the creation of strategic plans for the development of the city and mental health does not appear in a simple linear relationship, a certain positive effect on relational health can be seen in Figure 2. Our hypothesis is that participation in planning can stimulate the emergence of binding, bridging and bonding social capital. Convergence of opinions and attitudes, however, does not appear, at first glance, to be a prerequisite for reducing the level of depression and increasing the mental health of the population.

As part of community engagement, we want to shed a light on the relationship between community engagement or community belonging and health. As part of the survey, respondents were asked to indicate the types of communities (and specific examples) in which they participate in their activities; that is, whether they literally live a community way of life. Three types of communities were mostly considered: (1) active communities of neighbours (22.51% of respondents take a part), (2) informal communities of active friends with specific interests (31.83% of respondents take a part) and (3) communities formed within active NGOs (24.12% of respondents take a part).

Participation in the activities and projects of active local communities does not generally promote mental health (Figure 3). In the case of the relationship between participation in NGO-related community activities and mental health, a relatively significant negative relationship was even found. Almost 65% of the respondents participating in the activities of these communities described themselves as "activists" in the survey. This means that

activists, volunteers and community members in NGOs have an increased probability of mental health problems. This may be related to their increased level of perception of problems and challenges, personal involvement in local development, active involvement in the value struggle in society and significant time burdens associated with more professional activities that NGO communities perform compared to communities of neighbours or active friends. However, that community engagement clearly supports increasing relational health is clearly demonstrated in Figure 2 in the case of all mentioned types of communities.

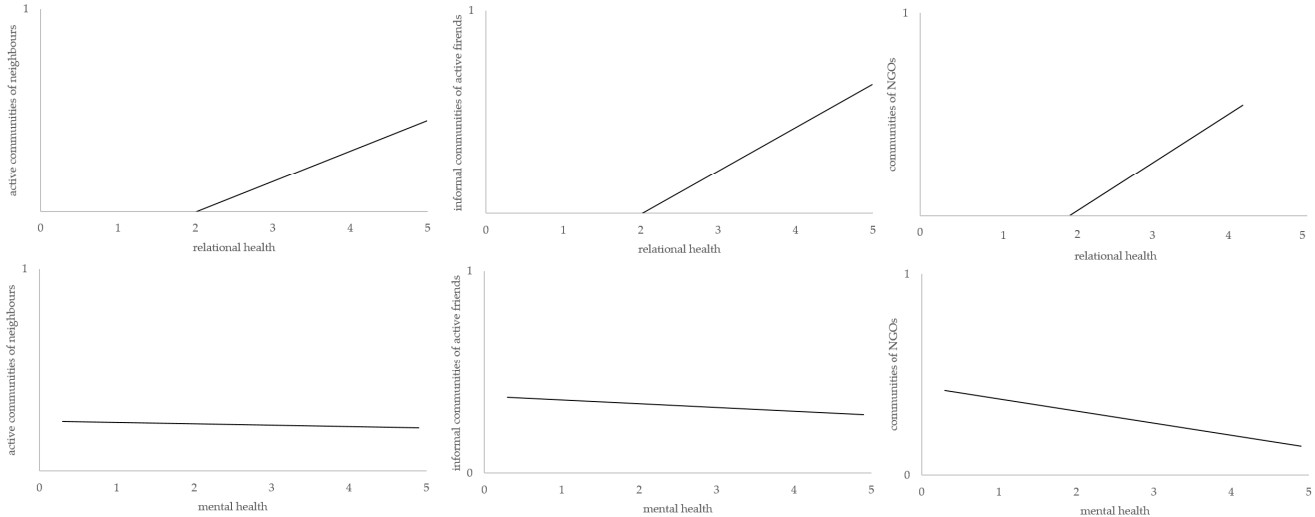

**Figure 3.** Prediction of linear relationships between several factors of community engagement, relational and mental health.

### 4.2. Results of Cross-Sectional Regression Model

Within the chapter, we present the results of the econometric analysis. The results are summarised in Table 4. Despite our previous descriptions in Section 4.1, we have to reject hypothesis H1 as a statistically significant but negative relationship was identified between the aggregate variable measuring the degree of political involvement in local politics and relational health. A high level of political participation from a complex point of view rather negatively affects the formation and maintenance of voluminous and high-quality interpersonal ties across families, friends and communities. However, we identified a positive relationship between political engagement and mental health, which means that political engagement contributes to increased life satisfaction and has a positive effect on depression. Therefore, we accept hypothesis H2.

On the contrary, social and community engagement contributes to the improvement of relational ties and relational health; therefore, we accept hypothesis H3. A statistically significant relationship between community engagement and mental health was not found, perhaps again due to the different relationship between mental health and the individual components of the aggregated variable. In general, community engagement in local development or community development gives the individual "meaning", a sense of fulfilment and a sense of value in society; however, on the other hand, it increases the inner pressure felt by the individual due to a broader understanding of the complex contexts of local problems, growing overload in terms of time-costs and inter-group social tensions. Therefore, we also reject hypothesis H4.

Similar to the case of the aggregated variable of political and community engagement, equivalent relationships were also identified between volunteering (measured by the average annual number of days during which the resident performed at least 1 h of volunteer work) or the dummy variable participation in the LDP (expressing whether the resident participated in the implementation of a project with a positive impact on the health and well-being of the population in local development) and relational or mental health.

**Table 4.** Results of cross-sectional regression model.

| | Relational Health | | | Mental Health | | |
|---|---|---|---|---|---|---|
| | *Model I* | *Model II* | *Model III* | *Model I* | *Model II* | *Model III* |
| AGE | −0.0142 ** | −0.0147 ** | −0.014 ** | −0.0235 *** | −0.0192 *** | −0.0197 *** |
| | (0.0055) | (0.0055) | (0.0055) | (0.0057) | (0.0057) | (0.0058) |
| GENDER | 0.0891 | 0.0976 | 0.0564 | 0.2951 ** | 0.3015 ** | 0.3176 ** |
| | (0.0775) | (0.0842) | (0.0873) | (0.1031) | (0.1054) | (0.1044) |
| FAITH | 0.2002 * | 0.0602 | 0.046 | −0.013 * | 0.0546 | 0.0616 |
| | (0.0945) | (0.1012) | (0.1021) | (0.1184) | (0.1168) | (0.1172) |
| FAMILY STATUS | 0.4719 *** | 0.4698 *** | 0.4382 *** | 0.5468 *** | 0.5311 *** | 0.5446 *** |
| | (0.0948) | (0.1037) | (0.1075) | (0.1313) | (0.1354) | (0.1365) |
| CHILDREN | −0.1333 *** | −0.1580 *** | −0.1531 *** | −0.0262 | −0.0259 | −0.0266 |
| | (0.0414) | (0.0443) | (0.0472) | (0.0588) | (0.0609) | (0.0614) |
| EDUCATION | 0.1669 ** | 0.2206 *** | 0.2657 *** | −0.0309 | 0.0109 | −0.0074 |
| | (0.0617) | (0.0608) | (0.0618) | (0.0844) | (0.0816) | (0.0793) |
| INCOME | 0.0012 | 0.0001 | 0.0001 | 0.0012 * | 0.0012 ** | 0.0012 ** |
| | (0.0001) | (0.0001) | (0.0001) | (0.0001) | (0.0001) | (0.0001) |
| FREE TIME | 0.0201 | 0.0158 | −0.0006 | 0.0235 * | 0.0162 | 0.0242 |
| | (0.019) | (0.0202) | (0.0199) | (0.0281) | (0.0298) | (0.0276) |
| POLITICAL_ENG | −0.1758 *** | | | 0.1722 ** | | |
| | (0.0504) | | | (0.0610) | | |
| COMMUNITY_ENG | 0.4133 *** | | | −0.0785 | | |
| | (0.0427) | | | (0.0588) | | |
| VOLUNTEERSHIP | | 0.2360 *** | | | −0.0752 | |
| | | (0.0345) | | | (0.0460) | |
| PARTICIP_LDP | | | 0.542 *** | | | −0.1343 |
| | | | (0.1121) | | | (0.1290) |
| const | 2.4797 ** | 2.6679 *** | 2.6499 *** | 3.5123 *** | 3.3788 | 3.3673 |
| | (0.2959) | (0.3007) | (0.306) | (0.3973) | (0.3947) | (0.3986) |
| no. obs | 311 | 311 | 311 | 311 | 311 | 311 |
| Breusch-Pagan hettest | 10.96 ** | 10.71 ** | 7.54 * | 4.99 * | 3.59 * | 5.98 * |
| VIF mean | 1.69 | 1.69 | 1.67 | 1.64 | 1.67 | 1.64 |
| linktest hat | 1.9729 *** | 2.4363 ** | 1.3286 ** | 0.4702 ** | 2.9009 ** | 2.4833 ** |
| linktest hat_sq | −0.1384 | −0.2192 | −0.0474 | 0.0783 | −0.2902 | −0.2266 |

All the estimates are robust due to problem of heteroscedasticity; data normalized using z-scoring, statistical significance on levels: * $p < 0.05$, ** $p < 0.01$, *** $p < 0.001$; standard errors in brackets.

Other results showed that relational and mental health decreases from the "middle-age" categories with further age increase and that women are more at risk than men from the point of view of mental health. Moreover, in the case of one of the models, we found a certain moderate positive effect of faith on relational health and a negative effect on mental health. Both concepts of health are increased in the case of married people, but with the increase in the number of children in the case of a given city, relational health decreased. We also identified that individuals with higher education generate richer and better-quality interpersonal ties. The assumption that higher income leads to a reduction in depressive symptoms was also confirmed. In conclusion, we state that we also found a certain positive relationship between the average amount of free time and the aggregate variable of mental health, which we consider a meaningful result.

## 5. Discussion

This case study at the level of a selected small, medium-sized city sheds light on relationships between civic engagement, understood as a mix of community and political engagement activities, and chosen aspects of population health. In line with Chandra et al. [4], we argue that civic engagement activities and processes can significantly support the health

of members of the local community, but our results do not correlate completely with previous studies.

The assumptions of relatively high levels of participation [11,12] in civic activities in the context of a small, medium-sized city have been fulfilled. Almost 48% of the population is involved in the life of communities of neighbours, communities of active friends or communities around actors of the non-profit sector. A relatively high proportion of respondents also participate in political engagement activities. These results show that the communities of small and medium-sized cities in Slovakia were quickly transforming and adapting to new possibilities connected with the development of civic rights and freedoms after the fall of socialism. Most authors agree that the suppression of civil society and the suppression of democratic values related to participation in public affairs during socialism de facto eroded the fundamental foundations of civil society [42–44]. The often-discussed experience of the population with collective temporary work supported the development of communities of small rural settlements rather than cities [42]. In the cases of cities in post-socialist countries, according to the opinions of other authors, civic engagement was renewed just slowly due to declining social trust after 1989 and worsening macroeconomic conditions [45], as pro-social behaviours are usually linked to positive social outcomes [46].

Concerning the results of the model, n contrary to other studies [4,9,13,21], taking into account the entire population over 15 years of age, we found no relationship between community engagement and mental health. However, this no longer applies when the aggregated indicator "community engagement" is decomposed into individual components. In the case of people living according to a "community-based" way of life, within groups of activists or citizens participating on activities of non-profit organisations, this relationship can even be negative. Respondents to open-ended questions pointed to the fact that the civil sector substitutes a large volume of missing public services in the conditions of the city. At the same time, the critical mass of highly active managers in the civil sector is small in the context of a city of this size category, which could explain why a high level of community engagement can also lead to an increase in the rate of depressive symptoms or burn-out. However, in the case of political engagement, in the same way as Wray-Lake et al. [21], we found a positive relationship with self-perceived mental health. In this case, the majority of the components of the aggregated variable contributed to the resulting relationship; depressive symptoms are mitigated by active participation in elections and participation in local council activities, as well as participation in protests, boycotts, petitions or through satisfaction found in discussions in online communities. The hypothesis of Liang and West [26] that relational health increases psychological distress was not confirmed in our case as we did not identify interdependencies.

As several authors suppose, civic engagement also affects relational health [34]. Community engagement and community activities contribute significantly to building bonds and improving overall relational health. However, political engagement, on the contrary, led to a decrease in the degree of relational health, which is probably a consequence of the high level of local society polarisation and the fact that the local government represented a government of activists at the time of the research. To understand the results, it may be important that at the time of the research, the local government had the support of more than 50% of population. When decomposing the aggregated variable "political participation" into individual components, it was found that it was participation in online protests and physical protests against the government of activists with support from the majority of the population that significantly contributed to the reduction of self-perceived relational health. Simply put, relational health is negatively affected by the political participation of opponents in the form of protests, which are mostly attended by residents with demands for more extreme solutions in city conditions. Considering the composition of the indicators for the aggregate variable, it can be hypothesised that conflicting attitudes towards local politics even affect the family ties of individuals.

Indirectly, our results also pointed to the potential impacts of community engagement on the collective responsibility for health equity [5] and mainly on addressing unhealthy conditions in communities [6] within open answers. We found that up to 43.7% of respondents directly participated, in the last 3 years, in health and well-being projects with potential impacts both on physical and mental health. Up to 20.2% of respondents participated in projects in their communities oriented towards improving access to fresh food and healthy eating habits awareness, 32.7% of respondents were engaged in activities aimed at reducing pollution and environmental burdens in the city, 37.6% of respondents participated in projects aimed at the development of community gatherings and meaningful leisure time with the often direct goal of supporting the mental health of vulnerable groups, 37.4% of respondents participated in the development of sports and exercise activities for residents and 24.4% of respondents in the sample participated in projects to support public health and develop community health and social services, field work and care of the sick. However, most of these residents participated in most types of projects at the same time.

As for the managerial implications, local policy makers and planners should consider the importance of civic engagement and the formation of grassroots communities for strengthening mental and relational health. The study encourages the integration of health as a shared value in strategic documents of local developments and local schemes and programs to support the activities of active communities. The NGOs and other non-formal active local community movements can supplement the missing activities and services of local governments in the area of public health and healthy lifestyle support. Increasing the breadth of participation in local development planning and involvement in local policymaking can similarly lead to an increase in the relational and mental health of the population. The results of the study point to the key importance of building cohesive active communities at the level of neighbourhoods, while the integration of citizens within these communities brings the greatest contribution to the reduction of depressive symptoms.

The main limitations of the study stem from the use of cross-section data and the inability to systematically collect this data for a longitudinal study. Our sample of respondents should be about 20 participants larger to perfectly meet the criteria for sample representativeness. We also consider the slightly reduced share of the age category of the population over 60 in the sample to be a limitation. At the same time, the 10-questions frameworks can only be considered as a certain indicative approach to the investigation of relational and mental health based on a limited spectrum of symptoms. Detailed data on the physical health of individuals were not available within the dataset. The COVID-19 pandemic, restricting the movement of the population and causing feelings of fear and uncertainty [47], affected civic engagement.

## 6. Conclusions

In the study, using a sample of the population living in a small, medium-sized city, we demonstrated the positive effects of the population's community engagement, including the positive effects of volunteering or the active participation of the population in health and well-being projects, on relational health. Our results have several theoretical implications. Especially due to specific local contextual conditions (significant opinion and political polarity and attitudes towards the non-profit sector and activism), the negative impact of political participation on relational health was found. On the contrary, the growth of the level of involvement in local politics contributed to the reduction of depressive symptoms, while a relationship between social involvement and self-perceived mental health measured by an aggregated variable was not found.

**Author Contributions:** Conceptualization, M.H. and P.M.; methodology, M.H. and K.M.; validation, K.M. and S.B.; investigation, M.H., P.M. and K.M.; resources, P.M. and S.B.; writing—original draft preparation, M.H., P.M., S.B. and K.M.; writing—review and editing, M.H. and P.M.; visualization, M.H.; supervision, M.H. and P.M.; project administration, K.M.; funding acquisition, K.M. All authors haveread and agreed to the published version of the manuscript.

**Funding:** This project has received funding from the European Union's Horizon 2020 research and innovation programme under grant agreement No. 869227 (IN-HABIT—INclusive Health And wellBeing In small and medium size ciTies).

**Institutional Review Board Statement:** Not applicable.

**Informed Consent Statement:** Not applicable.

**Data Availability Statement:** Data are available at: https://bit.ly/411R1BY.

**Conflicts of Interest:** The authors declare no conflict of interest.

## Appendix A

**Table A1.** 10-question framework for investigation of relational health.

| 10 Question Framework for Investigation of Relational Health |
| --- |
| (1) I invest a lot of time in meeting new people |
| (2) I care about good relationships with friends |
| (3) I care about good relations with colleagues and neighbours |
| (4) I can rely on my friends |
| (5) I do some of my activities mainly due to intention of socialising |
| (6) I am satisfied with the intensity of contact with my family |
| (7) In my family, close relationships based on trust prevail |
| (8) My expectations about partner and children are fulfilled |
| (9) I feel a sense of belonging to different communities with similar interests |
| (10) I like to spend time with other people in the community, it fulfils me |

## Appendix B

**Table A2.** 10-question framework for investigation of mental health in context of depression.

| 10 Question Framework for Investigation of Mental Health |
| --- |
| (1) I was also bothered by things that usually don't bother me |
| (2) I had trouble concentrating on the things I was doing |
| (3) I often felt depressed |
| (4) I felt that everything I do in life is an effort |
| (5) I felt optimistic about my future |
| (6) I often felt fear |
| (7) I have been suffering from sleep problems |
| (8) I was feeling happy |
| (9) I felt lonely |
| (10) I had a problem to start my activities |

## Appendix C

**Table A3.** 10-question framework for investigation of local political participation level.

| 10 Question Framework for Measuring Local Political Participation |
| --- |
| (1) I participate in the creation of territorial and development plans of the city |
| (2) I participate in the activities of city council commissions |
| (3) I participate in city council meetings |
| (4) I actively participate in the elections of self-governing representatives |
| (5) I participate passively in the elections of self-governing representatives |
| (6) In the past I worked or volunteered for a political party or trade unions |
| (7) I participate in public meetings and discussions with local government officials |
| (8) I participate in internet discussions about local policy |
| (9) I addressed a representative of the local government with a comment, complaint, or proposal |
| (10) I participate in physical or online protests, petitions or boycotts |

## Appendix D

**Table A4.** 10-question framework for investigation of civic engagement.

| **10 Question Framework for Measuring Local Community Engagement** |
| --- |
| (1) I am interested in the problems of the local communities or specific groups |
| (2) I am engaged in communities of specific interests (e.g., sports club, initiative of cyclists) |
| (3) I am involved in activities and projects in the neighbourhood |
| (4) In addition to life in a family, I live a community-based way of life |
| (5) I am part of active virtual communities |
| (6) I participated in projects with benefit to the public |
| (7) I participated in activities and solutions to support inclusion |
| (8) I provided 2% of the tax to NGO, participated on fundraising, or made a donation to NGO |
| (9) I implemented projects of NGO, or volunteered for NGO |
| (10) I define myself as a person who actively fights for social change |

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
