# Peer review of "Does Civic Engagement Support Relational and Mental Health of Urban Population?"

_societies, doi:10.3390/soc13020046_

Round 1

Reviewer 1 Report

The article "Does civic engagement support relational and mental health of urban population?" is interesting. The authors analyze civic engagement and its relationship with mental health. The subject matter is relevant and the approach is correct.

However, this paper presents some problems that hinder its proper understanding. For this reason, I consider that some modifications should be made. I specify these changes below:

1. The introduction talks about "civic engagement" but the explanation of the concept is given in the following section. To help the reader, I suggest the first sentence of paragraph 2 become the first sentence of the introduction. This is simply a suggestion; of course, the authors can vary this option according to their tastes.

2. In line 89 it is stated that "In many empirical studies..." but none of them are cited. It would be necessary to cite some works on this. Something similar happens in line 104, so I think it is necessary to include more citations in this line.

3. In Section 2 three important concepts are mentioned: "civic engagement", "mental health" (line 122) and "relational health" (line 146). But these three concepts are not well related. Civic engagement" is related to "well-being" but the other two concepts seem to be isolated. I suggest that an effort be made to relate the three elements well, since they will be of great importance later on. In the discussion, this relationship becomes clearer.

4. In the discussion, an effort should be made to compare, briefly, the results obtained with other surrounding countries.

5. The article presents some formatting errors. At the end of the tables there are explanatory notes that should indicate that they are indeed explanatory notes and separate them from the following paragraph. In addition, the bibliographical references should be revised to put them correctly. Finally, in-text citations need to be changed. This is what is indicated in the journal's rules:

“In the text, reference numbers should be placed in square brackets [ ], and placed before the punctuation; for example [1], [1–3] or [1,3]. For embedded citations in the text with pagination, use both parentheses and brackets to indicate the reference number and page numbers; for example [5] (p. 10). or [6] (pp. 101–105).

Author Response

Dear reviewer,

thank you for the review and constructive comments. We would like to answer questions/comments one by one.

  1. we shortened and reformulated first paragraph in Introduction section in order to draw the reader more quickly into the issue,
  2. not only in line 89, but in several places of the manuscript, we found similar situations where several authors are mentioned but not cited. We have resolved this issue,
  3. concerning comment no. 3, we must state that the Theoretical background section was not properly organized. Also in accordance with the requirements of other reviewers, we decided to first introduce the concepts of civic engagement, mental health and relational health and only then clarify the conclusions about identified mutual dependencies in empirical studies
  4. at this point we must note that the investigation of such relationships in empirical-deductive studies in the surrounding countries has not been recorded by us, we discuss the results of the authors mainly in Western Europe and US, where attention is paid to the topi,c
  5. there were indeed many formatting errors in the text. Explanatory notes are just under Table 1 and 5, and since we are not sure about the journal requirements in this regard, we distinguish them as most often default by italics. However, we believe that we will be guided when proofreading with an editor. At this stage, we have already used brackets to indicate citations. When sending manuscripts for review, we do not do it intentionally, as the manuscript goes through many changes and each time their order must be changed / marked anew.

In addition to the above, the manuscript went through the English editing service. Many typos were removed, redundant information in various parts was reduced. We moved the implications and limitations to the discussion.

We hope that the readability of the manuscript has increased significantly.

Thank you once again for your effort to contribute to improving the quality of the manuscript.

Authors.

Reviewer 2 Report

I found this research interesting and my comments are mainly formal.

Ll. 114-115 This statement is very interesting and probably true but a reference is needed.

L. 127 The dot is lacking.

L. 140-142 I could not understand why this study confirms the role of community engagement in public health improvement.

L. 316 I did not understand how optimism could drag down mental health.

L. 355 Sure about behavioral science, not sure about medicine, which I would not mention.

Ll. 397-405 Probably this part is better suited for the methodological section.

L. 442 I would not say "logically".

L. 455 I would make the text more concise by starting the discussion section directly from "In line with Chandra..."

Ll. 465-467 I find this statement circular.

Ll. 553-562 I think that the limitations should be explained in the final part of the discuss section.

Overall, the text should be summarized and made more concise to avoid redundant sentences and considerations, especially between the introduction and the theory background.

Extensive language revision is required.

Author Response

Thank you very much for your time and willingness to provide us with constructive feedback on the manuscript. In our answer, we will reflect on individual notes / comments on individual shortcomings.

First of all, several situations were identified when the sentence did not make proper sense, or the citation was missing, as was also pointed out in specific cases in your review (e.g. L. 114-115; L. 140-142; L. 355). We assessed each of the given situations and either reformulated the sentence, reduced it, or used a different term.

A very serious mistake was the incorrect statement in line L. 316 ("I did not understand how optimism could drag down mental health"). It is actually the "absence of feelings of optimism about the future" that contributed to the decrease in the final mental health score.

Also, we agree that lines Ll. 397-405 (first paragraph of the analysis results) contained either repetitive information or technical information that belongs to the Objectives and Methodology section.

Similarly, we left only one "introductory sentence" at the beginning of the discussion and reduced the repetition of goals and methodological approach.

We also agree that limitations and implications should be part of the discussion, so we have moved this information to the end of the discussion section.

We re-read the text and reduced the circular statements. We identified a number of sentences that required changes. We must state that the Theoretical Background section was not properly organized. Also in accordance with the requirements of other reviewers, we decided to first introduce the concepts of civic engagement, mental health and relational health and only then describe the conclusions about identified mutual dependencies in empirical studies. Several cited sources of literature have been reduced and some, on the contrary, have been added.

We used the services of an expert from a private company, who provided an English grammar spell-check of the entire text, identified the use of inappropriate expressions and a number of minor errors in tenses, inappropriate use of commas, conjunctions, etc. We hope that the overall readability of the manuscript has been improved.

Thank you once again for your effort to contribute to improving the quality of the manuscript.

Authors.

Round 2

Reviewer 1 Report

The explanations that have been introduced in the new version help to better understand the text. Thus the readability of the manuscript has increased significantly. Some problems have also been readjusted and solved. The effort made has resulted in a quality final document. Congratulations!

Reviewer 2 Report

Dear Authors,

I found your reply convincing.

Although the most important passages have been addressed, English language can still be improved by making the text more concise, but I leave the final judgement about it to the editor.

Thank you